# GANDALF: DATA AUGMENTATION IS ALL YOU NEED FOR EXTREME CLASSIFICATION

## ABSTRACT

Extreme Multi-label Text Classification (XMC) involves learning a classifier that can assign an input with a subset of most relevant labels from millions of label choices. Recent works in this domain have increasingly focused on the problem setting with short-text input data, and labels endowed with short textual descriptions called label features. Short-text XMC with label features has found numerous applications in areas such as prediction of related searches, title-based product recommendation, bid-phrase suggestion, amongst others. In this paper, we propose *Gandalf*, a graph induced data augmentation based on label features, such that the generated data-points can supplement the training distribution. By exploiting the characteristics of the short-text XMC problem, it leverages the label features to construct valid training instances, and uses the label graph for generating the corresponding soft-label targets, hence effectively capturing the label-label correlations. While most recent advances (such as SIAMESEXML and ECLARE) in XMC have been algorithmic, mainly aimed towards developing novel deep-learning architectures, our data-centric augmentation approach is orthogonal to these methodologies. We demonstrate the generality and effectiveness of *Gandalf* by showing up to 30% relative improvements for 5 state-of-the-art algorithms across 4 benchmark datasets consisting of up to 1.3 million labels.

## 1 INTRODUCTION

*Extreme Multilabel Classification* (XMC) has found multiple applications in the domains of related searches (Jain et al., 2019), product recommendation (Medini et al., 2019), dynamic search advertising (Prabhu et al., 2018), etc. which require predicting the most relevant results that either frequently co-occur or are highly correlated with the given product instance or search query. In the XMC setting, these problems are often modelled through embedding-based retrieval-cum-ranking pipelines over millions of possible web pages/products/ad-phrases considered as labels.

**Nature of short-text XMC and Extreme class imbalance** Typically, in the tasks of related search prediction, bid-phrase suggestion, and related-product recommendation based on titles, the input data instance is in the form of a *short-text* query. These short-text instances (names or titles), on average, consist of only 3-8 words . In order to effectively model these scenarios, there has been an increasing focus on building encoders as part of deep learning pipelines that can capture the nuances of such short-text inputs (Dahiya et al., 2021b; Kharbanda et al., 2021).

The real world datasets in XMC are highly imbalanced towards popular or trending ad-phrases/products. Moreover, these datasets adhere to Zipf's law (Ye et al., 2020), i.e., most labels in these extremely large output spaces are tail labels, having very few ($< 5$) instances in a training set spanning hundreds of thousands data points (Tab : 1, Appendix). While there is already an insufficiency of training data, the short-text nature of training instances makes it even more challenging for the models to learn meaningful, non-overfitting encoded representations for tail words and labels.

**Frugal architectures and Label features** Due to the low latency requirements of XMC applications, most recent works are also focused on building lightweight and frugal architectures that can predict in milliseconds and scale up to millions of labels (Dahiya et al., 2021a). Despite being frugal in terms of number of layers/parameters in the network, these models are capable of fitting well enough on the training data, although their generalization to the test samples remains poor (Fig : 1a). Hence, creating deeper models for better representation learning is perhaps not optimal under this setting.

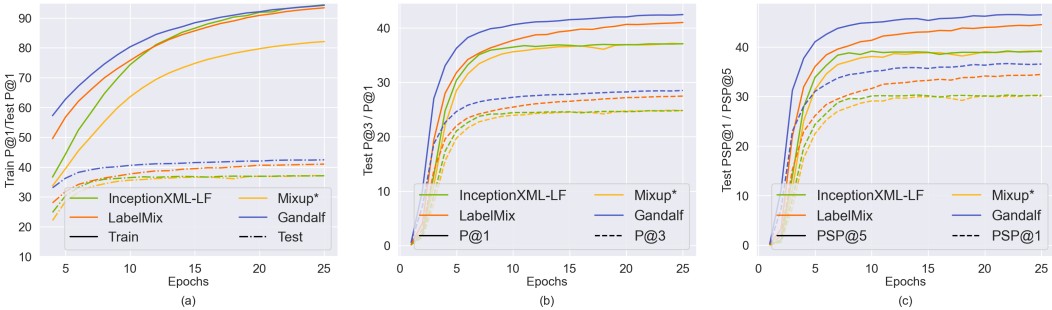

Figure 1: Effect of different data augmentations on INCEPTIONXML-LF over LF-AmazonTitles-131K dataset. (a) shows that a significant generalization gap exists between Train and Test P@1. However, remarkable improvements can be noted in (b) and (c) as a result of using the proposed data augmentation *Gandalf*. While text mixup (Chen et al., 2020) provides a regularization effect and is effective in reducing overfitting, our proposed alternative *LabelMix* baseline performs much better.

Recent works, however, make expensive architectural adjustments (Mittal et al., 2021a) to leverage the text associated with labels ("label features", discussed in §2) in order to improve generalization.

## 1.1 RELATED WORK: XMC WITH LABEL FEATURES

Earlier works in XMC primarily focused on problems consisting of entire long-text documents, consisting of hundreds of words/tokens, such as those encountered in tagging for Wikipedia (Babbar & Schölkopf, 2017; You et al., 2019). On the output side, the labels were identified by numeric IDs and hence devoid of any semantic meaning. Most works under this setting are aimed towards scaling up transformers as encoders for XMC tasks (Chang et al., 2020; Zhang et al., 2021).

By associating labels with their corresponding texts, which are in turn, product titles, document names or bid-phrases themselves, the contemporary application of XMC has gone beyond standard document tagging tasks. With the existence of label features, there exist three correlations that can be exploited for better representation learning: (i) query-label (ii) query-query and (iii) label-label correlations. Recent works have been successful in leveraging label features and pushing state-of-the-art by exploiting the first two correlations. For example, SIAMESEXML (Dahiya et al., 2021a) employs a siamese pre-training stage based on a contrastive learning objective between a data point and its label features optimizing negative log-likelihood loss. GALAXC (Saini et al., 2021) employs a graph convolutional network over a combined query-label bipartite graph. DECAF and ECLARE (Mittal et al., 2021a;b) make architectural additions to exploit higher order query-label correlations by extending the DeepXML pipeline to accommodate extra ASTEC-like encoders (Dahiya et al., 2021b).

In contrast to the recent algorithmic developments for short-text XMC with label features, and following the work of (Banko & Brill, 2001), which posits higher relevance of developing more training data as compared to choice of classifiers in small data regimes, we take a data-centric approach and focus on developing data augmentation techniques for short-text XMC.

## 1.2 CONTRIBUTIONS

This work has three-fold contributions:

- As our primary contribution, we propose *Gandalf* — **Gr**Aph i**N**duced **D**ata **A**ugmentation based on **L**abel **F**eatures — a simple data augmentation algorithm to efficiently leverage label features as valid training instances in XMC. Augmenting training data via *Gandalf* faciliates the core objective of short-text XMC by enabling the model to effectively capture label-label correlations in the latent space without the need of making architectural modifications,

- Empirically, we demonstrate the generality and effectiveness of *Gandalf* by showing up to 30% relative improvements in 5 state-of-the-art extreme classifiers across 4 public benchmark datasets. We show that by using *Gandalf*, methods which inherently do not leverage label features beat strong

baselines which either employ complicated training procedures (Dahiya et al., 2021a) or make heavy architectural modifications (Mittal et al., 2021a;b) to benefit by leveraging label features.

• In order to test *Gandalf* against a strong data-augmentation baseline, we propose *LabelMix* as an effective interpolation-based data augmentation baseline, which currently does not exist for short-text XMC. In the process of arriving at *LabelMix*, we also discuss the effectiveness of mixup (Zhang et al., 2018) and its variants and aim at answering "Can we extend mixup to feature-label extrapolation to guarantee a robust model behavior far away from the training data?", a question posed in (Zhang et al., 2018) as a future work.

## 2 WHAT EXACTLY ARE LABEL FEATURES?

To elaborate label features, we take examples relevant to our datasets (i) LF-WikiTitles-500K, where the model needs to predict the relevant categories, given only the title of a wikipedia page, and (ii) LF-AmazonTitles-131K, where given a product's name, model needs to recommend related products.

**Example 1:** For the wikipedia page *"2022 French presidential election"*, we have the available categories : *April 2022 events in France | 2022 French presidential election | 2022 elections in France | Presidential elections in France*. Further, a google search of the same query, leads to the following related searches - *French election 2022 - The Economist | French presidential election coverage on FRANCE 24 | Presidential Election 2022: A Euroclash Between a "Liberal... | French polls, trends and election news for France - POLITICO.eu*, amongst others.

**Example 2:** For *"Mario Kart: Double Dash!!"* on Amazon, we have available : *Super Smash Bros Melee | Super Mario Sunshine | Mario Party 7 | Super Mario Strikers* as the recommended products.

**Observations:** In view of these examples, one can affirm two important observations: (i) the short-text XMC problem indeed requires recommending similar items which are either highly correlated or co-occur frequently with the queried item, and (ii) the queried item and the corresponding label-features form an "equivalence class" and convey similar intent (Dahiya et al., 2021a). For example, a valid news headline search on a search engine should either result in a page mentioning the same headline or similar re-phrased headlines from other news media outlets (see Example 1). As a result, it can be argued that data instances are *interchangeable* with their respective labels' features.

## 3 GANDALF: DATA AUGMENTATION FOR EXTREME CLASSIFICATION

**Notation & Background** For training, we have available a multi-label dataset $\mathcal{D} = \{\{\mathbf{x}_i, \mathsf{y}_i\}_{i=1}^N, \{\mathbf{z}_l\}_{l=1}^L\}$[1] comprising of $N$ data points. Each $i \in [N]$ is associated with a small ground truth label set $\mathsf{y}_i \subset [L]$ from $L \sim 10^6$ possible labels. Further, $\mathbf{x}_i, \mathbf{z}_l \in \mathcal{X}$ denote the textual descriptions of the data point $i$ and the label $l$ which, in this setting, derive from the same vocabulary universe $\mathcal{V}$ (Dahiya et al., 2021a). The goal is to learn a parameterized function $f$ which maps each instance $\mathbf{x}_i$ to the vector of its true labels $\mathbf{y}_i \in \{0,1\}^L$ where $\mathbf{y}_{il} = 1 \Leftrightarrow l \in \mathsf{y}_i$.

A common strategy for handling this learning problem, called the *two towers* approach, is to map instances and labels into a common Euclidean space $\mathcal{E} = \mathbb{R}^d$, in which the relevance $s_l(\mathbf{x})$ of a label $l$ to an instance is scored using an inner product, $s_l(\mathbf{x}) = \langle \Phi(\mathbf{x}), \Psi(l) \rangle$. We call $\Phi(\mathbf{x})$ the encoding representation of the instance $\mathbf{x}$, and $\mathbf{w}_l := \Psi(l)$ the decoding representation of label $l$. If labels are featureless integers, then $\Psi$ turns into a simple table lookup. In our setting, $l$ is associated with features $\mathbf{z}_l$, so we identify $\Psi(l) = \Psi(\mathbf{z}_l)$.

The prediction function selects the $k$ highest-scoring labels, $f(\mathbf{x}) = \text{top}_k (\langle \Phi(\mathbf{x}), \Psi(\cdot) \rangle)$. Training is usually handled using the *one-vs-all* paradigm, which applies a binary loss function $\ell$ to each entry in the score vector. In practice, performing the sum over all labels for each instance is prohibitively expensive, so the sum is approximated by a shortlist of labels $\mathsf{S}(\mathbf{x}_i)$ that typically contains all the positive labels, and only those negative labels which are expected to be particularly challenging for classification (You et al., 2019; Dahiya et al., 2021b; Kharbanda et al., 2021), leading to

$$\mathcal{L}_\mathcal{D}[\Phi, \Psi] = \sum_{i=1}^N \sum_{l=1}^L \ell(\mathbf{y}_{il}, \langle \Phi(\mathbf{x}), \Psi(l) \rangle) \approx \sum_{i=1}^N \sum_{l \in \mathsf{S}(\mathbf{x}_i)} \ell(\mathbf{y}_{il}, \langle \Phi(\mathbf{x}), \Psi(l) \rangle). \quad (1)$$

[1] bold symbols $\mathbf{y}$ indicate vectors, captial letters $Y$ indicate random variables, and sans-serif $\mathsf{y}$ denotes a set

**Label Features as Data Points**    It is known that standard training on XMC datasets can easily lead to overfitting even with simple classifiers (Guo et al., 2019), which is confirmed for our setting in Fig : 1. To reduce the generalization gap, regularization needs to be applied to the label decoder $\Psi$, either explicitly as a new term in the loss function (Guo et al., 2019), or implicitly through the inductive biases of the network structure (Mittal et al., 2021a;b). Exploiting the interchangability of label and instance text, SIAMESEXML (Dahiya et al., 2021a) proposes to tie encoder and decoder together and require $\Psi(l) = \Phi(\mathbf{z}_l)$. While indeed yielding improved test performance, this approach has two drawbacks: Firstly, the condition $\Psi(l) = \Phi(\mathbf{z}_l)$ turns out to be too strong, and it has to allow for some fine-tuning corrections $\boldsymbol{\eta}_l$, yielding $\Psi(l) = \Phi(\mathbf{z}_l) + \boldsymbol{\eta}_l$. Secondly, training SIAMESEXML becomes a multi-staged process. Initially, a contrastive loss needs to be minimized, followed by fine-tuning with a classification objective.

Dahiya et al. (2021a) motivates its approach by postulating a self-annotation property (*Label Self Proximity*), which claims that a label $l$ is relevant to its own textual features with high probability, $\mathbb{P}[Y_l = 1 \mid X = \mathbf{z}_l] > 1 - \epsilon$ for some small $\epsilon \ll 1$. One natural question thus arises, *in a label space spanning* $\sim 10^6$ *labels, what are the other labels which annotate* $\mathbf{z}_l$, *when posed as a data point?* Therefore, to effectively augment the training set with $\mathbf{z}_l$ as a data point, we need to provide values for the other entries of the label vector $\mathbf{y}_l$. These labels should be sampled according to $\mathbf{y}_l \sim \mathbb{P}[\mathbf{Y} \mid X = \mathbf{z}_l]$, which means we need to find sensible approximations to the probabilities for the other labels $\mathbb{P}[Y_j = 1 \mid X = \mathbf{z}_l]$. When using the cross-entropy loss, sampling can be forgone by replacing the discrete labels $\mathbf{y}_l \in \{0, 1\}^L$ by soft labels $\mathbf{y}_l^{\text{soft}} = \mathbb{P}[\mathbf{Y} \mid X = \mathbf{z}_l]$.

**Exploiting Label Co-Occurrences**    In order to derive a model for $\mathbb{P}[Y_{l'} = 1 \mid X = \mathbf{z}_l]$, we can take inspiration from the GLAS regularizer (Guo et al., 2019). This regularizer tries to make the Gram matrix of the label embeddings $\langle \mathbf{w}_l, \mathbf{w}_{l'} \rangle$ reproduce the co-occurrence statistics of the labels $\mathbf{S}$,

$$\mathcal{R}_{\text{GLaS}}[\Psi] = L^{-2} \sum_{l=1}^{L} \sum_{l'=1}^{L} \left( \langle \mathbf{w}_l, \mathbf{w}_{l'} \rangle - S_{ll'} \right)^2 . \tag{2}$$

Here, $\mathbf{S}$ denotes the symmetrized conditional probabilities,

$$S_{ll'} := 0.5(\mathbb{P}[Y_l = 1 \mid Y_{l'} = 1] + \mathbb{P}[Y_{l'} = 1 \mid Y_l = 1]). \tag{3}$$

Plugging in $\mathbf{w}_l = \Psi(\mathbf{z}_l)$, this regularizer reaches its minimum if

$$\langle \Psi(\mathbf{z}_l), \Psi(\mathbf{z}_{l'}) \rangle = S_{ll'}. \tag{4}$$

By the self-proximity postulate, we can assume $\Psi(\mathbf{z}_l) \approx \Phi(\mathbf{z}_l)$. For a given augmented instance with target soft-label $(\mathbf{z}_l, y_{ll'}^{\text{soft}})$, the training will try to minimize $\ell(\langle \Phi(\mathbf{z}_l), \Psi(\mathbf{z}_{l'}) \rangle, y_{ll'}^{\text{soft}})$. To be consistent with equation 4, we therefore want to choose $y_{ll'}^{\text{soft}}$ such that $S_{ll'} = \arg \min \ell(\cdot, y_{ll'}^{\text{soft}})$. This is fulfilled for $y_{ll'}^{\text{soft}} = \sigma(S_{ll'})$ for $\ell$ being the binary cross-entropy, where $\sigma$ denotes the logistic function. If $\ell$ is the squared error, then the solution is even simpler, with $y_{ll'}^{\text{soft}} = S_{ll'}$.

For simplicity, and because of good empirical performance, we choose $y_{ll'}^{\text{soft}} = S_{ll'}$ even when training with cross-entropy loss. This results in the following, extended version of the self-proximity postulate:

**Postulate 1 (Soft-Labels for Label Features)** *Given a label $l$ with features $\mathbf{z}_l \in \mathcal{X}$, and a proxy for semantic similarity of labels $\mathbf{S}$, the labels features, when interpreted as an input instance, should result in predictions*

$$\mathbb{P}[Y_{l'} = 1 \mid X = \mathbf{z}_l] \approx S_{ll'}. \tag{5}$$

**Label Correlation Graph**    The label-similarity measure equation 3 used in the original GLaS regularizer uses only direct co-occurences of labels, which results in a noisy signal that does not capture higher-order label interdependencies. Therefore, we propose to replace it with the label correlation graph (LCG) as constructed in ECLARE. LCG $\in \mathbb{R}^{L \times L}$ is inferred by performing a random walk (with restarts) over a bipartite graph between input data instances and their corresponding ground-truth labels. Since entries in LCG are normalized and skewed in favor of tail labels, the LCG can be interpreted as a smoothed and regularized variant of the label co-occurrence matrix. More intuitively, (Mittal et al., 2021b) show that the LCG correctly identifies a set of semantically similar labels that either share tokens with the queried label, or co-occur frequently in the same context (for details, see Fig : 4 in Appendix A), thus making it a good candidate for a label-similarity measure.

While ECLARE uses the LCG to efficiently mine higher order query tail-label relations by augmenting the classifier $\Psi$ with graph information, we propose to leverage the graph weights (with an additional row-wise normalization to get values in range $[0, 1]$) as probabilistic soft labels for $\mathbf{z}_l$ as data instance. Further, to restrict the impact of noisy correlations in large output spaces (Babbar & Schölkopf, 2019), we empirically find it beneficial to threshold the soft labels obtained from LCG at $\delta = 0.1$ (uniformly for all datasets). The algorithmic procedure of the data augmentation via *Gandalf* is shown below :

---

### Algorithm 1: *Gandalf* Augmentation

```
1  # j - label index, Z - label feature token matrix
2  def Gandalf(j, Z, LCG, delta=0.1):
3      x = Z[j]
4      y = LCG[j, :] / LCG[j, j] #row-normalize LCG to obtain values in [0, 1]
5      y = numpy.where(y > delta, y, 0) #threshold noisy correlations
6      return (x, y)
```

---

**Capturing Label-label Correlations**   The models benefit from *Gandalf* in two ways: (i) from Fig. 3 it is evident that $\Phi(\mathbf{z}_l)$ does not exist in the vicinity of $\Phi(\mathbf{x}_i)$, for $l \in \mathbf{y}_i$, for either head or tail labels. Thus, *Gandalf* essentially expands the dataset by adding label features as data points which are far from training instances in $\mathcal{D}$ and, (ii) as labels are product names or document titles themselves, the new data points created through *Gandalf* essentially capture the apriori statistical correlations between products/documents that exist in the label space. As a result, the encoded representation of correlated labels, learnt by an underlying algorithm, are closer in the representation space. This especially benefits the tail labels which, more often than not, either get missed out during shortlisting or rank outside the desired top-k predictions. As shown in the experimental results (Table 2), the data points generated by *Gandalf*, indeed, lead to significant improvements for a suite of existing algorithms. It may be noted that apart from LCG, other sources of modeling correlations, such as those capturing global and local label correlations or a combination thereof, are also equally applicable (Huang & Zhou, 2012; Zhu et al., 2017).

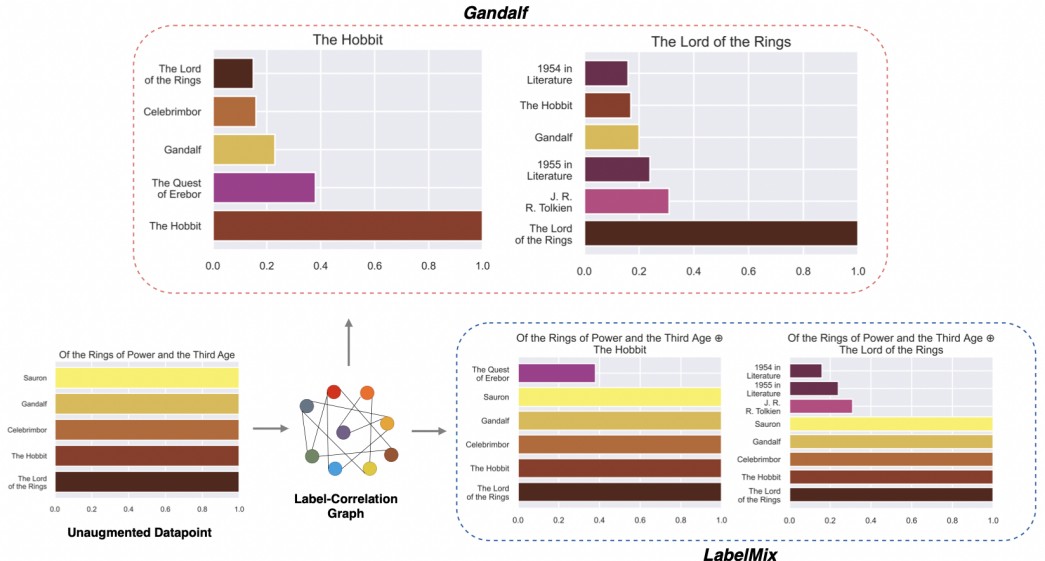

Figure 2: A pictorial representation of the proposed *Gandalf* and *LabelMix* strategies formed as per Alg : 1 and Eqn. 9. The title of each plot denotes the data point, the y-axis its labels and the x-axis their target values. We demonstrate our augmentations on the data point *Of the Rings of Power and the Third Age*, which is the final book in the Lord of the Rings(LOTR) series along with labels *The Hobbit* and *The Lord of the Rings*. Notably, the labels found through soft targets through the LCG are all related to the LOTR universe, with *J. R. R. Tolkien* being the author, *The Quest of Erebor* is a central plot line and, *Celebrimbor* and *Gandalf* are major characters. Beyond this, the soft targets also cover generic labels like *1954/55 in Literature*, which is the correct timeline for book release.

## 4 LABELMIX: QUERY-LABEL INTERPOLATION

Since the introduction of mixup for images (Zhang et al., 2018), approaches adapted for textual data (Guo et al., 2020; Chen et al., 2020) have also been proposed. Similar to Verma et al. (2019), these approaches propose to mix (interpolate) intermediate representations after $t$ layers $\{\phi_t(\mathbf{x}_i), \phi_t(\mathbf{x}_j)\}$ of the encoder $\Phi(\mathbf{x}) = \varphi_t(\phi_t(\mathbf{x}))$ along with the corresponding label vectors as:

$$\tilde{\phi}_t(\mathbf{x}_i, \mathbf{x}_j) := \lambda\phi_t(\mathbf{x}_i) + (1 - \lambda)\phi_t(\mathbf{x}_j); \quad \tilde{\mathbf{y}} := \lambda\mathbf{y}_i + (1 - \lambda)\mathbf{y}_j \tag{6}$$

where the mixing parameter $\lambda \in [0, 1]$ is sampled from $\mathrm{Beta}(\alpha, \alpha)$. The mixed latent representation $\tilde{\phi}_t$ is propagated through the rest of the encoder layers and the loss is calculated using the mixed label vector as $\ell(\langle\varphi_t(\tilde{\phi}_t), \Psi\rangle, \tilde{\mathbf{y}})$. However, we observe that while using such formulation of mixup does reduce overfitting by acting as a regularizer, it does not improve prediction performance on unseen data (refer to Mixup curves in Fig : 1). These observations are in line with (Chou et al., 2020), who argue that such formulation of $\tilde{\mathbf{y}}$ does not make sense under the imbalanced data regime and hence argue to create the mixed label vector to favor the minority class. In this section, we thus propose a new mixup technique - *LabelMix* as a strong data augmentation baseline for XMC, which favors tail labels and is more suitable for highly imbalanced problem as encountered in XMC.

Mixup techniques draw inspiration from *vicinal risk minimization*(VRM) (Chapelle et al., 2000). In VRM, a model is not trained to minimize the risk over the empirical distribution $\mathrm{d}\mathbb{P}_{\mathcal{D}}(\mathbf{x}, \mathbf{y}) = \frac{1}{n}\sum_{i=1}^{n}\delta_{\mathbf{x}_i}(\mathbf{x})\,\delta_{\mathbf{y}_i}(\mathbf{y})$, but instead over a smoothed out version $\mathbb{P}_{\mathrm{v}}$ which also comprises the vicinity of $\mathbf{x}$. The key task is then to determine what constitutes the vicinity of a data point.

**Query-Label Interpolation** In recommendation problems, formulated as short-text XMC tasks, works have focused on reducing distance between $\Phi(\mathbf{x}_i)$ and $\Phi(\mathbf{z}_l) \,\forall\, l \in \mathsf{y}_i$ in order to ensure high recall rate during the retrieval step and high efficiency while ranking the relevant labels (Mittal et al., 2021a;b; Saini et al., 2021). Thus, for the short-text XMC task at hand, we require the model to be invariant under a novel mixup transformation that relates more closely to the aforementioned recommendation objective. Since $\Phi(\mathbf{z}_l)$ is already expected to be in the vicinity of $\Phi(\mathbf{x}_i)$ and also exhibit such behaviour in a trained classifier (Fig : 3), the VRM perspective motivates to mix the encoded representations of a data point with one of its annotating label features as opposed to another data point in standard mixup formulations. We, therefore, propose to use a new definition of vicinity: given a data point $(\mathbf{x}_i, \mathsf{y}_i) \in \mathcal{D}$, its vicinity is given by $V(\mathbf{x}) \coloneqq \{\tilde{\phi}(\mathbf{x}_i, \mathbf{z}_l) : l \in \mathsf{y}_i\}$.

**Sampling Label for Mixup** In imbalanced data regimes, tail labels often have very few data points and thus it makes more sense to sample these labels more often. We thus use an instance-independent weight vector $\mathbf{r} \in \mathbb{R}^L$ (specifically, label frequency raised to the power 0.5 (Mikolov et al., 2013)), the probability of choosing $\mathbf{z}_l$ for interpolation from $\mathsf{y}_i$ is given by $\mathsf{y}_i \odot \mathbf{r}/\langle\mathsf{y}_i, \mathbf{r}\rangle$, where the denominator term ensures summation to unity.

While Dahiya et al. (2021a) employ a siamese contrastive loss between $\Phi(\mathbf{x}_i)$ and $\Phi(\mathbf{z}_l)$ *s.t.* $l \in \mathsf{y}_i$ in order to bring these closer in the latent space, we posit that an interpolation between these encoded representations in the latent space should result in an invariance i.e. keep the annotating labels unchanged. Intuitively, since the encoded representation of a data point is being mixed with that of one of its label's text, this should result in a *Label-Affirming Invariance*. More formally, we propose a novel postulate for query-label interpolation in shared embedding space:

**Postulate 2 (Label-Affirming Invariance)** *Let* $(\mathbf{x}, \mathsf{y})$ *be a training data point in* $\mathcal{D}$*, and* $l \in \mathsf{y}$ *be a label relevant to* $\mathbf{x}$*. Then the classifier should be invariant under mixup with* $\mathbf{z}_l$ *in the latent space*

$$\mathrm{top}_k(\langle\Phi(\mathbf{x}), \Psi\rangle) = \mathrm{top}_k(\langle\varphi_t(\tilde{\phi}_t(\mathbf{x}, \mathbf{z}_l)), \Psi\rangle) = \mathsf{y} ; \quad K = |\mathsf{y}| \tag{7}$$

Modifying Eqn. 6 using postulate 2 for a data point $(\mathbf{x}, \mathsf{y})$, we arrive at:

$$\tilde{\phi}_t(\mathbf{x}, \mathbf{z}_l) = \lambda\phi_t(\mathbf{x}) + (1 - \lambda)\phi_t(\mathbf{z}_l); \quad \tilde{\mathbf{y}} = \mathbf{y} \tag{8}$$

However, we find it empirically beneficial (ref. Tab : 3) to also accommodate for the label vector of $\mathbf{z}_l$ as proposed in postulate 1. This gives us *LabelMix*:

$$\tilde{\phi}_t(\mathbf{x}, \mathbf{z}_l) = \lambda\phi_t(\mathbf{x}) + (1 - \lambda)\phi_t(\mathbf{z}_l); \quad \tilde{\mathbf{y}} = \min(1, \mathbf{y} + \mathbf{y}_l^{\mathrm{soft}}) \tag{9}$$

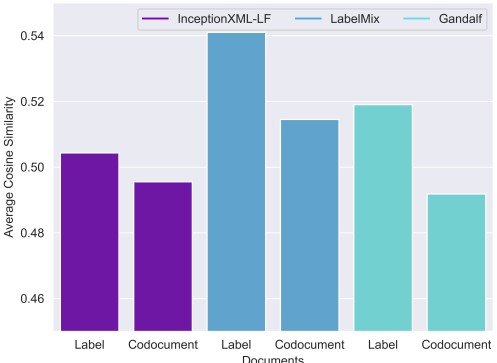

Figure 3: To obtain this plot, we take 50,000 product titles from LF-AmazonTitles-131K dataset and evaluate average cosine similarity between $\Phi(\mathbf{x}_i)$ and (i) $\Phi(\mathbf{z}_l)$ where $\mathbf{z}_l$ is a label feature of one of the annotating labels of $\mathbf{x}_i$, and (ii) $\Phi(\mathbf{x}_j)$, where $\mathbf{x}_i$ and $\mathbf{x}_j$ are "co-documents" i.e. share a label. Evidently, $\Phi(\mathbf{x}_i)$ is already closer to $\Phi(\mathbf{z}_l)$ in the embedding space as compared to $\Phi(\mathbf{x}_j)$ and this correlation increases by using the proposed augmentations.

Not only does *LabelMix* perform much better than standard mixup techniques (ref. Fig : 1), but also serves as a strong data augmentation baseline for short-text XMC, as shown in Tab : 2.

## 5 EXPERIMENTS & DISCUSSION

**Benchmarks, Baselines & Metrics** We benchmark our experiments on 4 standard public datasets, the details of which are mentioned in Tab : 1. To test the generality and effectiveness of our proposed *Gandalf*, we apply the augmentation across multiple state-of-the-art short-text extreme classifiers: (i) ASTEC, (ii) DECAF, (iii) ECLARE, and (iv) INCEPTIONXML. Additionally, we also compare against transformer-encoder based XR-Transformer (Zhang et al., 2021), and SiameseXML++. To compare *Gandalf* with conventional data augmentation approaches, we test it against *LabelMix* which serves as a strong mixup-based data augmentation baseline more suited for short-text XMC.

As an algorithmic contribution, we extend the INCEPTIONXML encoder to leverage label features in order to further the state-of-the-art on benchmark datasets and call it INCEPTIONXML-LF. For this, we augment the OVA classifier with additional label-text embeddings (LTE) and graph-augmented label embeddings (GALE) as done in (Mittal et al., 2021b). The implementation details and training strategy can be found in Appendix B. We measure the models' performance using standard metrics precision@k, denoted P@k, and its propensity-scored version PSP@k (Jain et al., 2016).

| Datasets | N | L | APpL | ALpP | AWpP |
|---|---|---|---|---|---|
| LF-AmazonTitles-131K | 294,805 | 131,073 | 5.15 | 2.29 | 6.92 |
| LF-WikiSeeAlsoTitles-320K | 693,082 | 312,330 | 4.67 | 2.11 | 3.01 |
| LF-WikiTitles-500K | 1,813,391 | 501,070 | 17.15 | 4.74 | 3.10 |
| LF-AmazonTitles-1.3M | 2,248,619 | 1,305,265 | 38.24 | 22.20 | 8.74 |

Table 1: Details of short-text benchmark datasets with label features. APpL stands for avg. points per label, ALpP stands for avg. labels per point and AWpP is the length i.e. avg. words per point.

### 5.1 MAIN RESULTS

We can make some key observations and develop strong insights not only about the short-text XMC problem with label features but also about specific dataset properties from Table 2. For example, the training on data points generated via *Gandalf* gives remarkable improvements on top of the base versions of existing algorithms especially on LF-AmazonTitles-131K and LF-WikiSeeAlsoTitles-320K where most labels have ∼5 training data points on average. In these low data regimes, *Gandalf* helps capture correlations which are not inherently captured by most existing models. In contrast, improvements on LF-WikiTitles-500K remain relatively mild where there is enough data per label for the models to be inherently able to capture these correlations.

***Gandalf*** With *Gandalf*, gains of up to 30% can be observed in case of ASTEC and INCEPTIONXML which, by default, do not leverage label features and yet perform at par with their LF-counterparts, i.e. DECAF and ECLARE, and INCEPTIONXML-LF across all datasets. While architectural modifications help capture higher order query-label relations and help model predict unseen labels better, they are computationally expensive, e.g. DECAF (having LTE) takes $\sim 2\times$ time to train while ECLARE (having both LTE & GALE) takes $\sim 3\times$ compared to its base model ASTEC. *Gandalf*-augmented

| Method | P@1 | P@3 | P@5 | PSP@1 | PSP@3 | PSP@5 | P@1 | P@3 | P@5 | PSP@1 | PSP@3 | PSP@5 |
|---|---|---|---|---|---|---|---|---|---|---|---|---|
| | LF-AmazonTitles-131K | | | | | | LF-AmazonTitles-1.3M | | | | | |
| AttentionXML | 32.25 | 21.70 | 15.61 | 23.97 | 28.60 | 32.57 | 45.04 | 39.71 | 36.25 | 15.97 | 19.90 | 22.54 |
| XR-Transformer | 38.10 | 25.57 | 18.32 | 28.86 | 34.85 | 39.59 | 50.14 | 44.07 | 39.98 | 20.06 | 24.85 | 27.79 |
| SiameseXML++ | 41.42 | **30.19** | 21.21 | 35.80 | 40.96 | 46.19 | 49.02 | 42.72 | 38.52 | 27.12 | 30.43 | 32.52 |
| Astec | 37.12 | 25.20 | 18.24 | 29.22 | 34.64 | 39.49 | 48.82 | 42.62 | 38.44 | 21.47 | 25.41 | 27.86 |
| + *LabelMix* | 37.95 | 25.65 | 18.59 | 29.91 | 35.58 | 40.63 | 49.56 | 44.01 | 39.57 | 21.69 | 26.13 | 28.05 |
| + *Gandalf* | 43.95 | 29.66 | 21.39 | 37.40 | 43.03 | 48.31 | 53.02 | 46.13 | 41.37 | 27.32 | 31.20 | 33.34 |
| Decaf | 38.40 | 25.84 | 18.65 | 30.85 | 36.44 | 41.42 | 50.67 | 44.49 | 40.35 | 22.07 | 26.54 | 29.30 |
| + *LabelMix* | 39.30 | 26.60 | 19.23 | 31.81 | 37.67 | 42.83 | 49.63 | 43.77 | 39.72 | 20.35 | 24.83 | 27.61 |
| + *Gandalf* | 42.43 | 28.96 | 20.90 | 35.22 | 42.12 | 47.61 | 53.02 | 46.65 | 42.25 | 25.47 | 30.14 | 32.83 |
| Eclare | 40.46 | 27.54 | 19.63 | 33.18 | 39.55 | 44.10 | 50.14 | 44.09 | 40.00 | 23.43 | 27.90 | 30.56 |
| + *LabelMix* | 40.34 | 27.54 | 19.96 | 33.48 | 39.74 | 45.11 | 50.55 | 44.50 | 40.38 | 23.40 | 27.92 | 30.58 |
| + *Gandalf* | 42.51 | 28.89 | 20.81 | 35.72 | 42.19 | 47.46 | **53.87** | **47.45** | **43.00** | 28.86 | 32.90 | 35.20 |
| InceptionXML | 36.79 | 24.94 | 17.95 | 28.50 | 34.15 | 38.79 | 48.21 | 42.47 | 38.59 | 20.72 | 24.94 | 27.52 |
| + *LabelMix* | 40.41 | 27.45 | 19.82 | 32.12 | 38.54 | 43.81 | 49.33 | 43.08 | 39.21 | 23.67 | 25.73 | 28.89 |
| + *Gandalf* | **44.67** | 30.00 | **21.50** | 37.98 | 43.83 | 48.93 | 50.80 | 44.54 | 40.25 | 25.49 | 29.42 | 31.59 |
| InceptionXML-LF | 40.74 | 27.24 | 19.57 | 34.52 | 39.40 | 44.13 | 49.01 | 42.97 | 39.46 | 24.56 | 28.37 | 31.67 |
| + *LabelMix* | 41.90 | 28.20 | 20.35 | 35.60 | 41.07 | 46.20 | 49.84 | 43.71 | 40.42 | 26.31 | 30.14 | 32.46 |
| + *Gandalf* | 43.84 | 29.59 | 21.30 | **38.22** | **43.90** | **49.03** | 52.91 | 47.23 | 42.84 | **30.02** | **33.18** | **35.56** |
| | LF-WikiSeeAlsoTitles-320K | | | | | | LF-WikiTitles-500K | | | | | |
| AttentionXML | 17.56 | 11.34 | 8.52 | 9.45 | 10.63 | 11.73 | 40.90 | 21.55 | 15.05 | 14.80 | 13.97 | 13.88 |
| SiameseXML++ | 31.97 | 21.43 | 16.24 | **26.82** | 28.42 | 30.36 | 42.08 | 22.80 | 16.01 | 23.53 | 21.64 | 21.41 |
| Astec | 22.72 | 15.12 | 11.43 | 13.69 | 15.81 | 17.50 | 44.40 | 24.69 | 17.49 | 18.31 | 18.25 | 18.56 |
| + *LabelMix* | 22.91 | 15.79 | 12.02 | 13.99 | 16.57 | 18.04 | 44.63 | 24.91 | 18.35 | 19.21 | 19.53 | 19.32 |
| + *Gandalf* | 31.10 | 21.54 | 16.53 | 23.60 | 26.48 | 28.80 | 45.24 | 25.45 | 18.57 | 21.72 | 20.99 | 21.16 |
| Decaf | 25.14 | 16.90 | 12.86 | 16.73 | 18.99 | 21.01 | 44.21 | 24.64 | 17.36 | 19.29 | 19.82 | 19.96 |
| + *LabelMix* | 26.55 | 18.04 | 13.75 | 17.86 | 20.46 | 22.61 | 44.22 | 24.47 | 17.3 | 21.37 | 20.72 | 20.69 |
| + *Gandalf* | 31.10 | 21.60 | 16.31 | 24.83 | 27.18 | 29.29 | 45.27 | 25.09 | 17.67 | 22.51 | 21.63 | 21.43 |
| Eclare | 29.35 | 19.83 | 15.05 | 22.01 | 24.23 | 26.27 | 44.36 | 24.29 | 16.91 | 21.58 | 20.39 | 19.84 |
| + *LabelMix* | 29.42 | 19.94 | 15.17 | 22.05 | 24.36 | 26.46 | 44.41 | 24.49 | 17.13 | 21.21 | 20.34 | 19.9 |
| + *Gandalf* | 31.33 | 21.40 | 16.31 | 24.83 | 27.18 | 29.29 | 45.12 | 24.45 | 17.05 | 24.22 | 21.41 | 20.55 |
| InceptionXML | 23.10 | 15.54 | 11.52 | 14.15 | 16.71 | 17.39 | 44.61 | 24.79 | 19.52 | 18.65 | 18.70 | 18.94 |
| + *LabelMix* | 25.16 | 17.03 | 12.97 | 16.11 | 18.72 | 20.76 | 44.85 | 24.91 | 19.73 | 19.37 | 18.98 | 19.56 |
| + *Gandalf* | 32.54 | 22.15 | 16.86 | 25.27 | 27.76 | 30.03 | 45.93 | 25.81 | **20.36** | 21.89 | 21.54 | 22.56 |
| InceptionXML-LF | 28.99 | 19.53 | 14.79 | 21.45 | 23.65 | 25.65 | 44.89 | 25.71 | 18.23 | 23.88 | 22.58 | 22.50 |
| + *LabelMix* | 29.68 | 20.16 | 15.32 | 22.24 | 24.69 | 26.80 | 45.64 | 26.35 | 18.78 | 24.09 | 22.98 | 23.00 |
| + *Gandalf* | **33.12** | **22.70** | **17.29** | 26.68 | **29.03** | **31.27** | **47.13** | **26.87** | 19.03 | **24.12** | **23.92** | **23.82** |

Table 2: Results showing the effectiveness and generality of *Gandalf* on state-of-the-art extreme classifiers.

base encoders, on the other hand, do not need to make any architectural modifications or employ complicated training pipelines to imbue necessary invariances.

***LabelMix*** While being effective in capturing query tail-label correlations, *LabelMix* can only imbue limited additional inductive bias into the model. ECLARE, on the other hand, is able to better capture these higher order correlations through its label graph-augmented classifier (GALE), and thus only gains trivially from *LabelMix*. DECAF gains non-trivially on both LF-AmazonTitles-131K and LF-WikiSeeAlsoTitles-320K datasets as it only encodes label text embeddings (LTE) in its classifier, which leaves out the scope to capture query tail-label correlations further. Similarly, INCEPTIONXML stands to gain significantly more from *LabelMix* compared to its LF-counterpart which also employs GALE. Notably, *LabelMix* works much better on INCEPTIONXML(-LF) than ECLARE because of their dynamic negative mining, which enables the augmentation to work more effectively.

***Gandalf* vs GALE** ECLARE leverages LCG to encode label-label correlations in $\mathbf{w}_l$ through GALE which helps the model improve prediction performance on new unseen labels. However, this only allows the classifier to distribute the loss gradient from a training instance $\{\mathbf{x}_i, \mathbf{y}_i\}$ across $\mathbf{y}_i$ and correlated labels as per LCG. This essentially captures higher order query-label correlations while not exploiting label-labels correlations in a way *Gandalf* does. Since the correlations learnt from GALE and *Gandalf* are independent of each other, we find ECLARE and INCEPTIONXML-LF, both of which employ GALE, to benefit off training on data points generated using *Gandalf*.

## 5.2 ABLATION STUDY

We try using *Gandalf* and *LabelMix* without soft-labels (*SL*) from LCG in Table 3, where *Gandalf w/o SL* is essentially equivalent to using label features as data points with self-annotation property

alone. However, that only helps the model learn label-to-words associations, like LTE in DECAF. Notably, soft-targets play an important role in enabling the encoder to intrinsically learn the label-label correlations Table 3 and imbue the necessary inductive bias in the models. For further analysis, we provide visualizations depicting differences in prediction performances obtained with and without by our proposed augmentations in Appendix B (Table 5).

| Method | P@1 | P@3 | P@5 | PSP@1 | PSP@3 | PSP@5 | P@1 | P@3 | P@5 | PSP@1 | PSP@3 | PSP@5 |
|---|---|---|---|---|---|---|---|---|---|---|---|---|
| | | | LF-AmazonTitles-131K | | | | | | LF-WikiSeeAlsoTitles-320K | | | |
| InceptionXML | 35.62 | 24.13 | 17.35 | 27.53 | 33.06 | 37.50 | 21.53 | 14.19 | 10.66 | 13.06 | 14.87 | 16.33 |
| +Synonym Replacement | 35.07 | 23.71 | 17.08 | 27.20 | 32.41 | 36.77 | 20.08 | 13.13 | 9.92 | 12.00 | 13.50 | 14.90 |
| + *LabelMix w/o SL* | 37.25 | 25.02 | 17.98 | 29.25 | 34.58 | 39.09 | 22.61 | 14.98 | 11.30 | 14.02 | 15.95 | 17.55 |
| + *LabelMix* | 39.05 | 26.52 | 19.15 | 30.98 | 37.20 | 42.26 | 23.90 | 16.10 | 12.28 | 15.20 | 17.60 | 19.56 |
| + *Gandalf w/o SL* | 37.59 | 25.25 | 18.18 | 30.75 | 35.54 | 40.06 | 24.43 | 16.16 | 12.15 | 16.89 | 18.45 | 20.02 |
| + *Gandalf* | 43.52 | 29.23 | 20.92 | 36.96 | 42.71 | 47.64 | 31.31 | 21.38 | 16.22 | 24.31 | 26.79 | 28.83 |

Table 3: Results demonstrating the effectiveness of using *Gandalf* soft-labels (denoted *SL*) and synonyms replacement on a single InceptionXML model.

## 6   OTHER RELATED WORK : DATA AUGMENTATION AND XMC

Architectural design choices are often complemented with data augmentation methodologies which have been found to be successful in imbuing necessary problem-specific invariances in the model, thereby improving model's generalization capability on unseen data. Textual augmentations in the discrete space such as making spelling errors (Xie et al., 2017), WordNet-based (Miller et al., 1990) replacement with synonyms (Kolomiyets et al., 2011; Li et al., 2017; Wang et al., 2018), text-fragment switch (Andreas, 2020), random insertion, swap and deletion as proposed in versions of EDA (Wei & Zou, 2019; Karimi et al., 2021) have been shown to bring some performance improvements (Coulombe, 2018). However, such transformations can lead to semantic inconsistency and illegibility and, thus decrease performance for classification tasks (Qiu et al., 2020; Anaby-Tavor et al., 2020).

More recent methods have tried filling these gaps in semantic consistency; (Zhao et al., 2022) improve upon EDA by converting the requirements of diversity and semantic consistency as a min-max optimization problem. Many methods leverage language models to suggest context-specific replacements for masked tokens either discretely via a single synonym (Kobayashi, 2018; Wu et al., 2019) or as a weighted sum of word embeddings of semantically similar words (Gao et al., 2019).

Even though the above approaches help mitigate semantic inconsistency to some extent, they are not able to preserve the annotating label, especially in low data regimes (Hu et al., 2019) where a major chunk of XMC data lies. These issues of semantic inconsistency and label distortion can be more explicit, particularly for short-text instances in XMC i.e. document titles or product names, where each word in the query has high correlation with the labels. Deletion or insertion of a word in the query could completely alter the search to either a more generalized or narrowed down one, or result in something with little sense. For example, changing the search query from *"Beats Wireless headphones"* to *"Beats Wireless headphones with microphone"* would lead to a filtered result. Furthermore, similar to label-altering random crops in images (which can be considered as the visual equivalent of word deletion) as pointed out by (Balestriero et al., 2022), altering the aforementioned query to remove or replace "Beats" with a synonym might lead to a result not having the intended brand in top 10 hits.

## 7   CONCLUSION

In this paper, we proposed *Gandalf*, a data augmentation strategy, which is particularly suited for short-text extreme classification. It not only eliminates the need for complicated training procedures in order to imbue inductive biases, but dramatic increase in prediction performance of state-of-the-art methods in this domain. Additionally, we also developed LabelMix, as a baseline data augmentation which is motivated from previous interpolation-based textual mixup techniques. It is expected that our treatment towards studying invariances in this domain will spur further data-centric research on designing other data augmentation methods to effectively replace architectural additions in order to leverage label features and achieve faster inference times.

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

## A    VISUALIZATIONS

The highly sparse nature of the XMC problem makes the LCG noisy. In order to reduce this noise from our soft targets, we threshold the correlation values at $\delta$, and quantify its effect by varying the parameter, as shown in Table 4. Additional visualizations capturing the label correlations and their first order-neighbors are shown in Figure 4. To better denote the impact of *Gandalf* on tail label prediction, we perform a quantile analysis by distributing the labels into 5 equi-voluminous bins based on the label frequency in the training data, as shown in Figure 5. Finally, the qualitative comparison of correctness of outputs generated by the baseline model, and those as a result of the proposed augmentations is shown in Table 5.

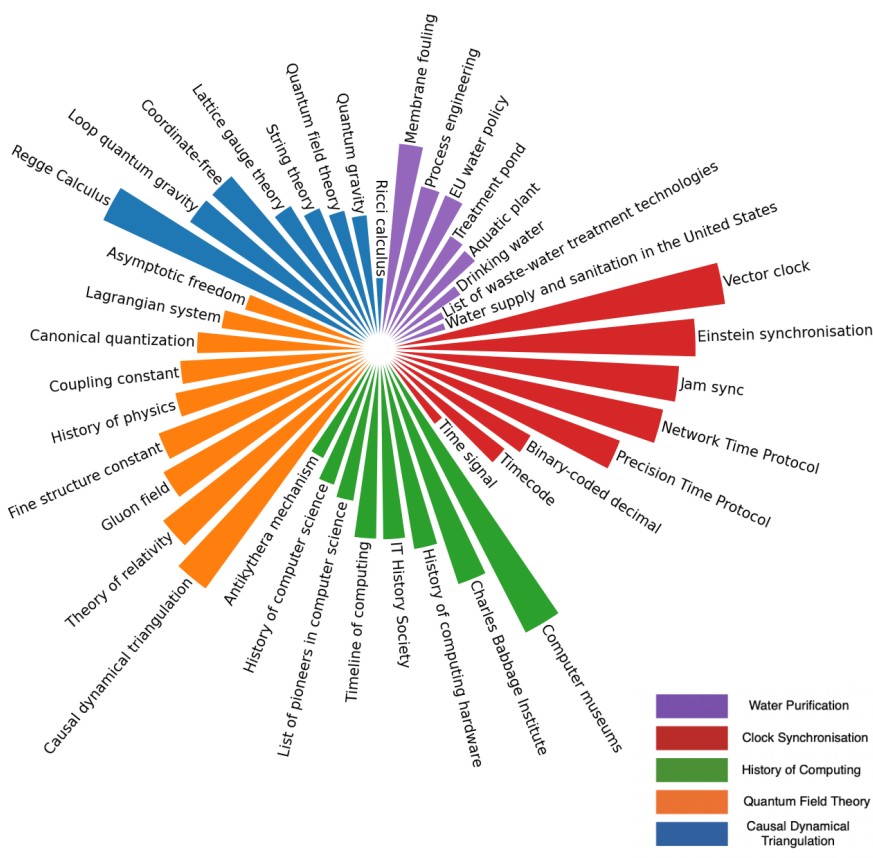

Figure 4: Correlations between labels and their first-order neighbours, as found by the LCG on the LF-WikiTitles-500K dataset. The legend shows the label in question, the bar chart shows the degree of correlation with its neighbouring labels. Correlated labels often share tokens with each other and/or may be used in the same context.

| | P@1 | P@3 | P@5 | PSP@1 | PSP@3 | PSP@5 | P@1 | P@3 | P@5 | PSP@1 | PSP@3 | PSP@5 |
|---|---|---|---|---|---|---|---|---|---|---|---|---|
| $\delta$ | | | | LF-AmazonTitles-131K | | | | | | LF-WikiSeeAlsoTitles-320K | | |
| 0.0 | 41.71 | 28.03 | 20.14 | 36.94 | 41.93 | 46.64 | 31.40 | 21.56 | 16.53 | 26.01 | 27.89 | 29.99 |
| 0.1 | **42.09** | **28.38** | **20.45** | **37.09** | **42.19** | **47.04** | **32.20** | **21.86** | **16.60** | **26.06** | **28.01** | **30.03** |
| 0.2 | 41.73 | 28.10 | 20.18 | 37.01 | 41.99 | 46.67 | 31.29 | 21.35 | 16.28 | 25.68 | 27.59 | 29.65 |
| 0.3 | 41.39 | 27.74 | 19.89 | 36.71 | 41.51 | 46.09 | 31.03 | 20.92 | 15.99 | 25.11 | 27.12 | 29.14 |

Table 4: Results demonstrating the sensitivity of *Gandalf* with respect to $\delta$, as defined in Algorithm 1. All experiments were performed on the InceptionXML-LF model, augmented with *Gandalf*. As shown, the empirical performance peaks at a $\delta$ value of 0.1 which is sufficient to suppresses the impact of noisy correlations.

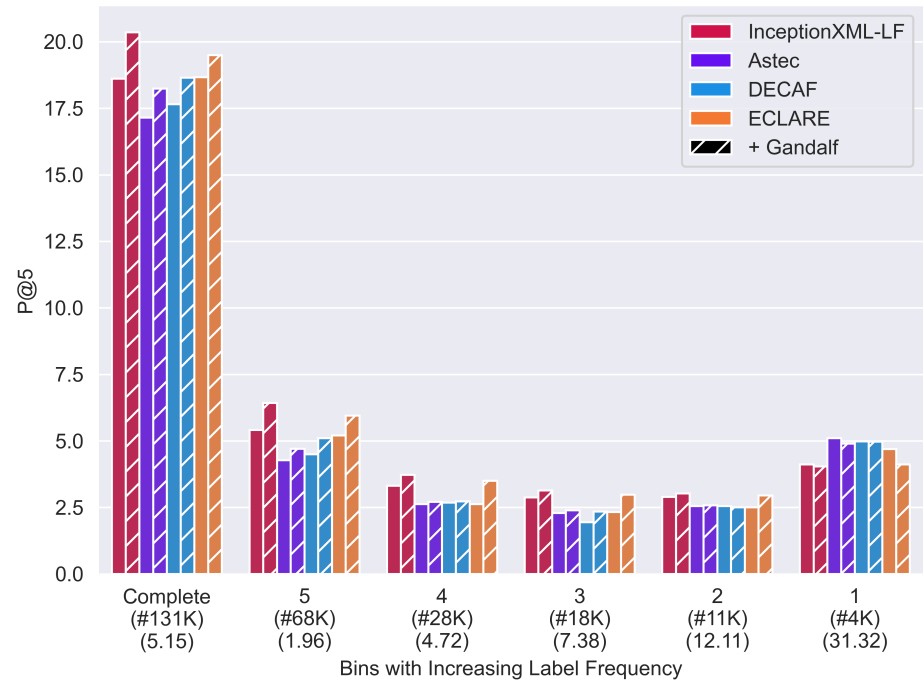

(a) Contributions to P@5 in LF-AmazonTitles-131K

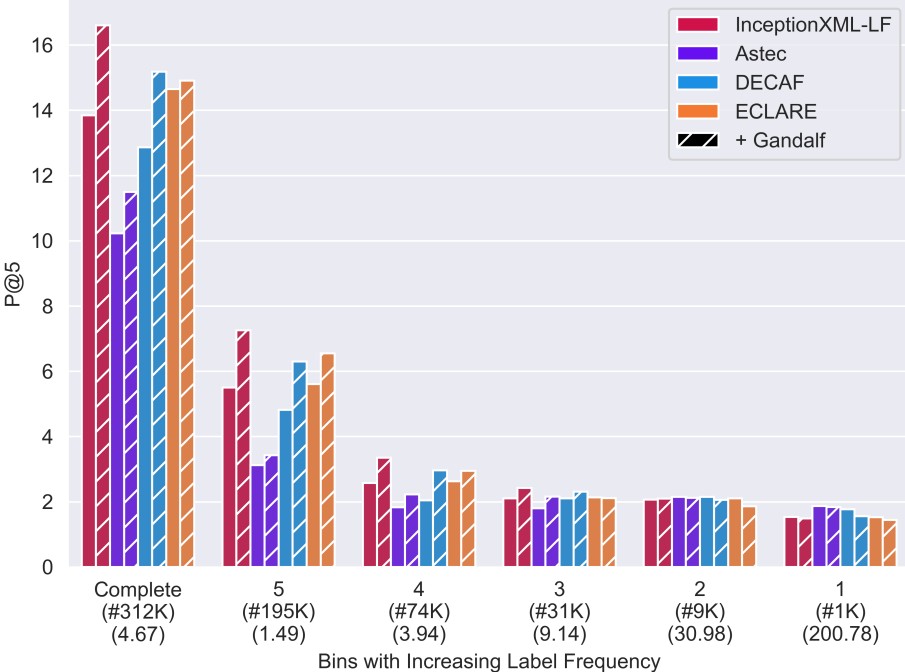

(b) Contributions to P@5 in LF-WikiSeeAlsoTitles-320K

Figure 5: Analysis demonstrating the effectiveness of *Gandalf* in improving performance over tail labels. For this graph, labels were divided into 5 equi-voluminous bins in increasing order of frequency. The graph shows contribution of each bin to P@5 on different datasets and short-text extreme classifiers.

| Method | Datapoint | Baseline Predictions | *LabelMix* Predictions | *Gandalf* Predictions |
|---|---|---|---|---|
| INCEPTIONXML-LF | | Pontryagin duality, Topological order, Topological quantum field theory, Topological quantum number, Quantum topology | Topological order, Algebraic group, Topological ring, Topological quantum field theory, Topological quantum number | Compact group, Haar measure, Lie group, Algebraic group, Topological ring |
| DECAF | Topological group | Topological quantum computer, Topological order, Topological quantum field theory, Topological quantum number, Quantum topology | Topological order, Algebraic group, Topological ring, Topological quantum field theory, Topological quantum number | Compact group, Haar measure, Lie group, Algebraic group, Topological ring |
| ECLARE | | Topological quantum computer, Topological order, Topological quantum field theory, Topological quantum number, Quantum topology | Topological order, Topological ring, Topological quantum field theory, Topological quantum number, Quantum topology | Compact group, Topological order, Lie group, Algebraic group, Topological ring |
| INCEPTIONXML-LF | | List of lighthouses in Scotland, List of Northern Lighthouse Board lighthouses, Oatcake, Communes of the Finistere department, Communes of the Cotes-d'Armor department | Oatcake, Oat milk, Rolled oats, List of oat diseases, Goboat | Oatcake, Oatmeal, Oat milk, Porridge, Rolled oats |
| DECAF | Oat | Oatcake, Oatmeal, Design for All (in ICT), Oatley Point Reserve, Oatley Pleasure Grounds | Oatcake, Oatmeal, Oat milk, Oatley Point Reserve, Oatley Pleasure Grounds | Oatcake, Oatmeal, Oat milk, Porridge, Rolled oats |
| ECLARE | | Oatmeal, Oat milk, Parks in Sydney, Oatley Point Reserve, Oatley Pleasure Grounds | Oatmeal, Rolled oats, McCann's Steel Cut Irish Oatmeal, Oatley Point Reserve, Oatley Pleasure Grounds | Oatcake, Porridge, Rolled oats, Oatley Point Reserve, Oatley Pleasure Grounds |
| INCEPTIONXML-LF | | Colorado metropolitan areas, Front Range Urban Corridor, Outline of Colorado, Index of Colorado-related articles, State of Colorado | Colorado metropolitan areas, Outline of Colorado, Index of Colorado-related articles, State of Colorado, Colorado counties | Colorado metropolitan areas, Outline of Colorado, Index of Colorado-related articles, Colorado cities and towns, Colorado counties |
| DECAF | Grand Lake, Colorado | Colorado metropolitan areas, Front Range Urban Corridor, State of Colorado, Colorado municipalities, National Register of Historic Places listings in Grand County, Colorado | Front Range Urban Corridor, Index of Colorado-related articles, National Register of Historic Places listings in Grand County, Colorado, Grand County, Colorado, List of lakes in Colorado | Outline of Colorado, State of Colorado, Colorado cities and towns, Colorado municipalities, Colorado counties |
| ECLARE | | State of Colorado, Colorado cities and towns, Colorado counties, National Register of Historic Places listings in Grand County, Colorado, Grand County, Colorado | Colorado metropolitan areas, State of Colorado, Colorado cities and towns, Colorado counties, Colorado census designated places | Outline of Colorado, Index of Colorado-related articles, State of Colorado, Colorado cities and towns, Colorado counties |
| INCEPTIONXML-LF | | Lunar Orbiter Image Recovery Project, Lunar Orbiter 3, Lunar Orbiter 5, Chinese Lunar Exploration Program, List of future lunar missions | Exploration of the Moon, List of missions to the Moon, Lunar Orbiter Image Recovery Project, Lunar Orbiter 3, Lunar Orbiter 5 | Surveyor program, Luna programme, Lunar Orbiter Image Recovery Project, Lunar Orbiter 3, Lunar Orbiter 5 |
| DECAF | Lunar Orbiter program | Exploration of the Moon, List of man-made objects on the Moon, Lunar Orbiter Image Recovery Project, Lunar Orbiter 3, Lunar Orbiter 5 | Exploration of the Moon, Lunar Orbiter program, Lunar Orbiter Image Recovery Project, Lunar Orbiter 3, Lunar Orbiter 5 | Exploration of the Moon, Apollo program, Surveyor program, Luna programme, Lunar Orbiter program |
| ECLARE | | Exploration of the Moon, Lunar Orbiter program, Lunar Orbiter Image Recovery Project, Lunar Orbiter 3, Lunar Orbiter 5 | Exploration of the Moon, Lunar Orbiter program, Lunar Orbiter Image Recovery Project, Lunar Orbiter 3, Lunar Orbiter 5 | Exploration of the Moon, Pioneer program, Surveyor program, Luna programme, Lunar Orbiter program |
| INCEPTIONXML-LF | | Royal Saudi Air Defense, Royal Saudi Strategic Missile Force, Saudi Royal Guard Regiment, Terrorism in Saudi Arabia, Capital punishment in Saudi Arabia | Saudi-led intervention in Bahrain, Royal Saudi Navy, Royal Saudi Air Defense, Royal Saudi Strategic Missile Force, Saudi Royal Guard Regiment | Military of Saudi Arabia, Royal Saudi Air Force, Royal Saudi Air Defense, Royal Saudi Strategic Missile Force, King Khalid Military City |
| DECAF | Armed Forces of Saudi Arabia | Saudi Arabian-led intervention in Yemen, Saudi-led intervention in Bahrain, Human rights in Saudi Arabia, Legal system of Saudi Arabia, Joint Chiefs of Staff (Saudi Arabia) | Saudi-led intervention in Bahrain, Saudi Arabia, Military of Saudi Arabia, Royal Saudi Strategic Missile Force, Saudi Arabian National Guard | Royal Saudi Air Force, Royal Saudi Navy, Royal Saudi Air Defense, Royal Saudi Strategic Missile Force, Saudi Arabian National Guard |
| ECLARE | | List of armed groups in the Syrian Civil War, Military of Saudi Arabia, Royal Saudi Strategic Missile Force, King Khalid Military City, Joint Chiefs of Staff (Saudi Arabia) | Military of Saudi Arabia, Royal Saudi Air Defense, King Khalid Military City, Saudi Royal Guard Regiment, List of rulers of Saudi Arabia | Military of Saudi Arabia, Royal Saudi Air Defense, Royal Saudi Strategic Missile Force, King Khalid Military City, Saudi Royal Guard Regiment |

Table 5: Prediction examples of different datapoints from the LF-WikiSeeAlsoTitles-320K dataset. Labels indicate mispredictions. It may be noted that queries with even just a single word, like *"Oat"*, which predicts unrelated labels in the case of the baseline prediction, gets all the labels right with the addition of *Gandalf*. Furthermore, even mispredictions get closer when our data augmentation strategy is introduced.

# B   INCEPTIONXML-LF

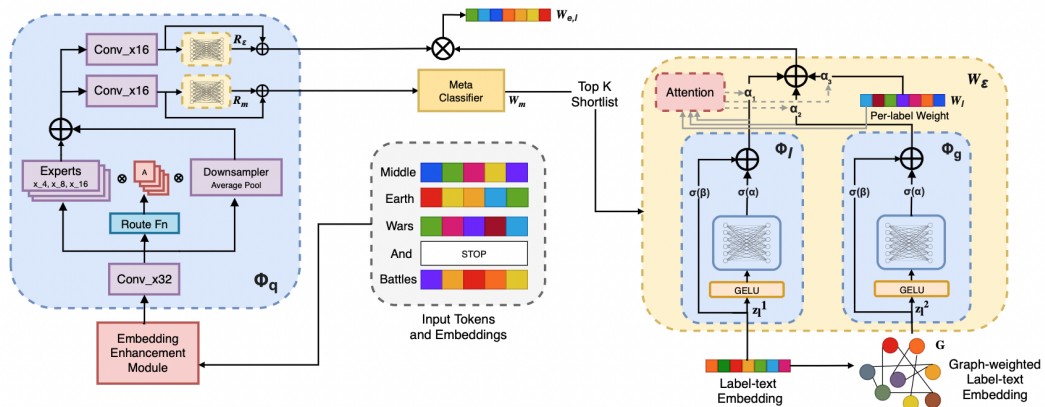

Figure 6: INCEPTIONXML-LF. The improved Inception Module along with instance attention is shown in detail. Changes to the INCEPTIONXML framework using the ECLARE classifier are also shown.

**Model Outlook:**   Short-text queries are encoded by a modified InceptionXML encoder, which encodes an input query $\mathbf{x}_i$ using an encoder $\Phi_q := (E, \theta)$ parameterised by $E$ and $\theta$, where $E$ denotes a $D$-dimensional embedding layer of $\mathbb{R}^{\mathcal{V} \times D}$ for vocabulary tokens $\mathcal{V} = [t_1, t_2, \ldots, t_V]$ and $\theta$ denotes the parameters of the embedding enhancement and the inception module respectively. Alongside $\Phi_q$, INCEPTIONXML-LF learns two frugal ASTEC-like Dahiya et al. (2021b) encoders, one each as a label-text encoder $\Phi_l := \{E, \mathcal{R}\}$ and a graph augmented encoder $\Phi_g := \{E, \mathcal{R}\}$. Here, $\mathcal{R}$ denotes the parameters of a fully connected layer bounded by a spectral norm and the embedding layer $E$ is shared between all $\Phi_q$, $\Phi_l$ and $\Phi_g$ for joint query-label word embedding learning. Further, an attention module $\mathcal{A}$, meta-classifier $\mathcal{W}_m$ and an extreme classifier $\mathcal{W}_e$ are also learnt together with the encoders. Next, we specify the details of all components of INCEPTIONXML-LF.

## B.1   INSTANCE-ATTENTION IN QUERY ENCODER

We make two improvements to the inception module INCEPTIONXML for better efficiency. Firstly, in the inception module, the activation maps from the first convolution layer are concatenated before passing them onto the second convolution layer. To make this more computationally efficient, we replace this "inception-like" setting with a "mixture of expert" setting Yang et al. (2019). Specifically, a route function is added that produces dynamic weights for each instance to perform a dynamic element-wise weighted sum of activation maps of each filter. Along with the three convolutional experts, we also add an average pool as a down sampling residual connection to ensure better gradient flow across the encoder.

Second, we decouple the second convolution layer to have one each for the meta and extreme classification tasks.

## B.2   DYNAMIC HARD NEGATIVE MINING

Training one-vs-all (OvA) label classifiers becomes infeasible in the XMC setting where we have hundreds of thousands or even millions of labels. To mitigate this problem, the final prediction or loss calculation is done on a shortlist of size $\sqrt{L}$ comprising of only hard-negatives label. This mechanism helps reduce complexity of XMC from an intractable $O(NDL)$ to a computationally feasible $O(ND\sqrt{L})$ problem. INCEPTIONXML-LF inherits the synchronized hard negative mining framework as used in the INCEPTIONXML. Specifically, the encoded meta representation is passed through the meta-classifier which predicts the top-K relevant label clusters per input query. All labels present in the top-K shortlisted label clusters then form the hard negative label shortlist for the extreme task. This allows for progressively harder labels to get shortlisted per short-text query as the training proceeds and the encoder learns better representations.

### B.3 LABEL-TEXT AND LCG AUGMENTED CLASSIFIERS

INCEPTIONXML-LF's extreme classifier weight vectors $\mathcal{W}_e$ comprise of 3 weights, as in Mittal et al. (2021b). Specifically, the weight vectors are a result of an attention-based sum of (i) label-text embeddings, created through $\Phi_l$, (ii) graph augmented label embeddings, created through graph encoder $\Phi_g$ and, (iii) randomly initialized per-label independent weights $\mathbf{w}_l$.

As shown in Fig. 6, we first obtain label-text embeddings as $\mathbf{z}_l^1 = E \cdot \mathbf{z}_l^0$, where $\mathbf{z}_l^0$ are the TF-IDF weights of label feature corresponding to label $l$. Next, we use the label correlation graph $\mathbf{G}$ to create the graph-weighted label-text embeddings $\mathbf{z}_l^2 = \sum_{m \in [L]} \mathbf{G}_{lm} \cdot \mathbf{z}_l^0$ to capture higher order query-tail label correlations. $\mathbf{z}_l^1$ and $\mathbf{z}_l^2$ are then passed into the frugal encoders $\Phi_l$ and $\Phi_g$ respectively. These encoders comprise only of a residual connection across a fully connected layer as $\alpha \cdot \mathcal{R} \cdot \mathcal{G}(\tilde{z}_l) + \beta \cdot \tilde{z}_l$, where $\tilde{z}_l = \{\mathbf{z}_l^1, \mathbf{z}_l^2\}$, $\mathcal{G}$ represents GELU activation and $\alpha$ and $\beta$ are learned weights. Finally, the per-label weight vectors for the extreme task are obtained as

$$\mathcal{W}_{e,l} = \mathcal{A}(\mathbf{z}_l^1, \mathbf{z}_l^2, \mathbf{w}_l) = \alpha^1 \cdot \mathbf{z}_l^1 + \alpha^2 \cdot \mathbf{z}_l^2 + \alpha^3 \cdot \mathbf{w}_l$$

where $\mathcal{A}$ is the attention block and $\alpha^{\{1,2,3\}}$ are the dynamic attention weights produced by the attention block.

### B.4 TWO-PHASED TRAINING

**Motivation:** We find there to be a mismatch in the training objectives in DeepXML-based approaches like ASTEC, DECAF and ECLARE which first train their word embeddings on meta-labels in Phase I and then transfer these learnt embeddings for classification over extreme fine-grained labels in Phase III Dahiya et al. (2021b). Thus, in our two-phased training for INCEPTIONXML-LF, we keep our training objective same for both phases. Note that, in INCEPTIONXML-LF the word embeddings are always learnt on labels instead of meta-labels or label clusters and we only augment our extreme classifier weight vectors $\mathcal{W}_e$ with label-text embeddings and LCG weighted label embeddings. We keep the meta-classifier $\mathcal{W}_m$ as a standard randomly initialized classification layer.

**Phase I:** In the first phase, we initialize the embedding layer $E$ with pre-trained GloVe embeddings Pennington et al. (2014), the residual layer $\mathcal{R}$ in $\Phi_l$ and $\Phi_g$ is initialized to identity and the rest of the model comprising of $\Phi_q$, $\mathcal{W}_m$ and $\mathcal{A}$ is randomly initialized. The model is then trained end-to-end but without using free weight vectors $\mathbf{w}_l$ in the extreme classifier $\mathcal{W}_e$. This set up implies that $\mathcal{W}_e$ only consists of weights tied to $E$ through $\Phi_l$ and $\Phi_g$ which allows for efficient joint learning of query-label word embeddings Mittal et al. (2021a) in the absence of free weight vectors. Model training in this phase follows the INCEPTIONXML+ pipeline as described in Kharbanda et al. (2021) without detaching any gradients to the extreme classifier for the first few epochs. In this phase, the final per-label score is given by:

$$P_l = \mathcal{A}(\Phi_l(\mathbf{z}_l^1), \ \Phi_g(\mathbf{z}_l^2)) \cdot \Phi_q(x)$$

**Phase II:** In this phase, we first refine our clusters based on the jointly learnt word embeddings. Specifically, we recluster the labels using the dense $\mathbf{z}_l^1$ representations instead of using their sparse PIFA representations Chang et al. (2020) and consequently reinitialize $\mathcal{W}_m$. We repeat the Phase I training, but this time the formulation of $\mathcal{W}_e$ also includes $\mathbf{w}_l$ which are initialised with the updated $\mathbf{z}_l^1$ as well. Here, the final per-label score is given by:

$$P_l = \mathcal{A}(\Phi_l(\mathbf{z}_l^1), \ \Phi_g(\mathbf{z}_l^2), \ \mathbf{w}_l) \cdot \Phi_q(x)$$

