# OpenReview forum: "Gandalf : Data Augmentation is all you need for Extreme Classification"
_ICLR.cc/2023/Conference — Submitted to ICLR 2023_

### Official Review · Reviewer_xmDr · 2022-10-24

**Confidence:** 4
**Correctness:** 3
**Technical Novelty And Significance:** 3
**Empirical Novelty And Significance:** 3
**Recommendation:** 6

**Clarity, Quality, Novelty And Reproducibility:**

- Clarity: The writing is clear
- Novelty: The idea of data augmentations are novel for XMC community
- Reproducibility: No code provided, hence no data point to verify the reproducibility


**Strength And Weaknesses:**

*Strength*
- The data augmentation techniques (Gandalf and LabelMix) are novel in XMC
- Promising performance gain

*Weaknesses*
- Lack of ablation study to back the effectiveness of certain component in Gandalf

**Summary Of The Paper:**

This paper studied extreme multi-label classification problems where labels short text descriptions are available. The author proposed Gandalf, a data-augmentation technique that augmented the label text as input data, and used the label correlation graph as its soft-labels. The proposed Gandalf technique was applied to various XMC models, and showed consistent improvement on four XMC benchmarks.

**Summary Of The Review:**

This work presented two novel data augmentation techniques, Gandalf and LabelMix, for the XMC problems. When applied to various downstream XMC  models, both Gandalf and LabelMix demonstrated consistent improvement over the baseline. Overall writing and presentation also are clear to follow.

The paper quality can be improved if the following technique questions are addressed.
- Choice of label-to-label graph: Gandalf with Co-occurrence graph instead of LCG?
- Sensitivity of hyper-parameters such as delta in Algo 1 and the design choice of y_tilde in Eq(9) versus Eq(8).
- Does Gandalf enjoys larger performance gain on tail labels (e.g., label count < 5)? Can you plot the performance gain in y-axis, and the groups of label freq in x-axis?
- I suppose Gandalf and LabelMix are complementary to each other? Can we apply both?
- Can you apply Gandalf/LabelMix to two-tower models (e.g., DPR/ANCE) and do the evaluation on both XMC and IR benchmark datasets?

---

> ### Author Response · Authors · 2022-11-19
> **response to Reviewer xmDr**
>
> 1. “Choice of label-to-label graph: Gandalf with Co-occurrence graph instead of LCG?”
>
> Purely statistical label co-occurrence matrix are extremely sparse and highly unreliable in XMC (ProXML [1], Section 4). We, therefore, propose to leverage LCG (as proposed in ECLARE) which is a smoothed version of the label co-occurrence matrix skewed in favor of tail labels. We mention this in the Label Correlation Graph paragraph in Section 3.
>
> -------
>
> 2. “Design choice of y_tilde in Eq(9) versus Eq(8)”
>
> As we want the encoded representations of both the data point and its label’s text to be in close proximity in the embedding space, we tried out training the new data point on the complete objective (hard labels + soft labels of label feature generated through Gandalf) instead of following the Label Affirming Invariance derived from the results of SiameseXML. As LabelMix was our secondary contribution, we skipped a discussion on this due to space constraints. However, in a later version of the paper, we shall add this discussion.
>
> -------
>
> 3. Sensitivity of hyper-parameters such as delta in Algo 1 and performance of Gandalf on tail labels.
>
> * We updated our Appendix A to include ablation on value of delta. This can be found on page 14.
> * Yes, huge improvements in the PSP metric (which denotes performance on tail labels) can be witnessed from Table 2.
> * We also include the requested graph in Appendix A. This can be found on page 15.
>
> -------
>
> 4. “I suppose Gandalf and LabelMix are complementary to each other? Can we apply both?”
>
> We tried applying both together, however, it results in a slight decrement in results obtained purely from using Gandalf. We believe that the learning objectives of both the augmentations are not as complementary to each other as one’s intuition might suggest. On one hand, Gandalf trains only label features with soft labels and data points with their ground truth hard labels. On the other hand, LabelMix suggests training a mix of a data point and its labels’ text on a combination of hard and soft targets. We believe LabelMix’s objective acts as a regularizer and reduces model’s learning capacity during training, thus results in a dip in performance as compared to only using Gandalf. Again, in the interest of space we skipped this discussion. We shall add this ablation and relevant discussion in a future version of the paper.
>
> ---------
> 5. “Can you apply Gandalf/LabelMix to two-tower models (e.g., DPR/ANCE) and do the evaluation on both XMC and IR benchmark datasets?”
>
> Thanks for pointing this out. In fact, applying Gandalf i.e. LCG based soft-targets in contrastive learning is part of our ongoing research work. As this task is non-trivial and requires optimal batch sampling for effective training, we wish to submit the same as a future work independent of this paper.

---

### Official Review · Reviewer_k6DK · 2022-10-25

**Confidence:** 5
**Correctness:** 2
**Technical Novelty And Significance:** 2
**Empirical Novelty And Significance:** 2
**Recommendation:** 3

**Clarity, Quality, Novelty And Reproducibility:**

* For clarity, the paper needs to be polished, especially for the organization. The flow is not easy to follow, especially in the part “proposing the baseline LabelMix”.
* The idea is interesting and kind-of novel, but the technical quality of experiments are unsatisfactory.
* For reproducibility, the authors provide some implementation details.


**Details Of Ethics Concerns:**

I found the paper is still concurrently under review in ACL ARR https://openreview.net/forum?id=hPG9DW8shI

**Strength And Weaknesses:**

 Strengths
* Simple and effective approaches that potentially can apply to arbitrary methods.
* Publicly available benchmark datasets for reproducibility.

Weaknesses
* The authors “propose” LabelMix as a method as the “baseline”, but the more authentic approach should be to compare with published state-of-the-art works in data augmentation for XMC tasks.
* Experiments only cover datasets with only short text (i.e., titles), which are comparatively limited in the real-world settings. The authors should consider conducting experiments on more comprehensive datasets with general information, such as LF-Amazon-131K, LF-WikiSeeAlso-320K, and LF-Wikipedia-500K.
* Although the proposed techniques are potentially able to be applied to arbitrary methods, some methods like AttentionXML, XR-Transformer, and SiameseXML++ are not experimented with either LabelMix or Gandalf. Especially when vanilla XR-Transformer and SiameseXML++ outperform LabelMix and Gandalf in certain cases, I cannot see why the authors do not demonstrate that the proposed techniques can further improve the best performance.
* The proposed techniques require label features, but in real-world settings, it usually lacks label features.


**Summary Of The Paper:**

In this paper, the authors propose to conduct graph induction for data augmentation for extreme classification based on label features. Specifically, they capture label-label correlations based on a label-correlation graph and use label features as data points. They also leverage label features to have query-label interpolation and then sample labels for mixing up augmented labels. The experiments based on four extreme classification benchmarks with label features demonstrate that the proposed method can enhance some extreme classification models.

**Summary Of The Review:**

In sum, I would recommend “3: reject, not good enough” as there are so many flaws in the experiments.

---

> ### Author Response · Authors · 2022-11-11
> **response to Reviewer k6DK**
>
> 1. “The authors “propose” LabelMix as a method as the “baseline”, but the more authentic approach should be to compare with published state-of-the-art works in data augmentation for XMC tasks.”
>
> We propose LabelMix as a “data augmentation baseline for XMC”. Can you please suggest which are the state-of-the-art works for data augmentation in XMC tasks that we may have left out? To the best of our knowledge, there are no other papers in this line of work. Indeed, in the absence of any existing methods, we are proposing LabelMix in this paper as a strong baseline. This has already been mentioned in the LabelMix section. Furthermore, we compare against existing NLP based approaches for augmentation such as bert-based synonym replacement, the results of which are provided in Table 3, and reviewed in Section 6.
> __________
>
> 2. “Experiments only cover datasets with only short text (i.e., titles), which are comparatively limited in the real-world settings. The authors should consider conducting experiments on more comprehensive datasets with general information, such as LF-Amazon-131K, LF-WikiSeeAlso-320K, and LF-Wikipedia-500K” & “The proposed techniques require label features, but in real-world settings, it usually lacks label features”
>
> * To the contrary, label text/features are available in most real-world commercial application of XMC like prediction/matching settings such as Wikipedia, query-to-product matching in recommendation engines, dynamic search ads, related search prediction in search engines, and so on in the form document name, product name, ad keyword etc.
> * Further, we would argue that it is short-text setting which gives the XMC framework its versatility to be applicable in the many aforementioned commercial applications. As a result, the focus of recent works in XMC ([DECAF](http://manikvarma.org/pubs/mittal21-main.pdf), [ECLARE](http://manikvarma.org/pubs/mittal21b.pdf), [GalaXC](http://manikvarma.org/pubs/saini21.pdf), [SiameseXML](http://proceedings.mlr.press/v139/dahiya21a/dahiya21a.pdf)) has shifted heavily on both short-text and LF datasets, which can be seen in depiction of results in [Extreme Classification Repository](http://manikvarma.org/downloads/XC/XMLRepository.html).
> * Moreover, *as emphasized in the abstract of the paper, our focus in this work is on short-text XMC problems*, where we can exploit the symmetry between input instances and the corresponding output labels of the dataset to achieve the data augmentation. This symmetry is no longer available in long-text inputs datasets mentioned by the reviewer.
>  _________
>
> 3. “Although the proposed techniques are potentially able to be applied to arbitrary methods, some methods like AttentionXML, XR-Transformer, and SiameseXML++ are not experimented with either LabelMix or Gandalf. Especially when vanilla XR-Transformer and SiameseXML++ outperform LabelMix and Gandalf in certain cases”
>
> * We would like to highlight that we show the effectiveness and generality of our data augmentations on a variety of state-of-the-art *short-test* extreme classifiers, which is the focus of this paper. Improvements on SiameseXML++ via Gandalf would be added to the rebuttal revision version of the paper.
> * The proposed augmentation scheme can be applied to AttentionXML and XR-Transformer, these results will be added in an updated version of the paper. These were not given in Table 2 of the paper for two reasons : (1) These works focussed on long-text problems, and perform worse compared to the much stronger (short-text) baselines used in the paper such as ECLARE, and DECAF, and (2) space constraints.
> * The main purpose of the paper is not to compare algorithms against each other but to demonstrate the proposed augmentation is general enough to be applicable to a wide range of algorithms. However, if one still wants an algorithmic comparison, then the existing baselines should be compared to the algorithmic contribution of this paper, namely InceptionXML-LF. It can be observed that the InceptionXML-LF with Gandalf data augmentation works better than all existing algorithms with or without augmentations.
>
> _____________
>
> 4. "I found the paper is still concurrently under review in ACL ARR https://openreview.net/forum?id=hPG9DW8shI"
>
> We appreciate the reviewer for going to the lengths of checking for dual submission. However, we would like to highlight that we submitted this paper to the July cycle of ACL ARR. An ARR cycle is concluded when the paper receives at least 3 reviews and a meta review and is no longer considered under submission. The ARR review cycle for our paper concluded on 22nd September 2022.

---

### Official Review · Reviewer_g2YZ · 2022-10-25

**Confidence:** 4
**Correctness:** 3
**Technical Novelty And Significance:** 2
**Empirical Novelty And Significance:** Not applicable
**Recommendation:** 3

**Clarity, Quality, Novelty And Reproducibility:**

The paper is well presented and easy to follow. However, as mentioned above, the main method is a straightforward extension of existing work thus the novelty is limited. I have no concerns for reproducibility as long as the authors plan to release the code.

**Strength And Weaknesses:**

**Strength**

* The proposed method Gandalf shows relatively large improvement over 5 XMC models across 4 different XMC-LF dataset.

**Weaknesses**

* The proposed method, Gandalf, is rather straightforward. The core idea seems to be just applying normalization and some thresholding on top of the LCG constructed in ECLARE.
* It is not clear to me how the Gandalf and LabelMix are related to each other. They seem to be two independent data augmentation tricks and I failed to see the necessity in introducing LabelMix in this paper.
* In Section the authors claim that LCG is better than Label co-occurrence matrix but both are not as good as Gandalf which adopts a heuristic thresholding after row-wise normalization. However, I didn't see any empirical study that compares these three settings. It would be more convincing if there are some label-to-label augmentation baselines in the comparison.





**Summary Of The Paper:**

The paper present two data augmentation strategies, Gandalf and LabelMix, that adds label features or the instance-label interpolations as training instances in extreme multi-label classification (XMC) training. Empirical results show that the proposed Gandalf method is able improve the performance of various XMC models on the benchmark dataset where label features are present.

**Summary Of The Review:**

The paper present two independent data augmentation tricks for the XMC-LF applications. Apart from the limited novelty concerns mentioned above, I think the topic of this paper is more related to a data mining venue rather than ICLR.

---

> ### Author Response · Authors · 2022-11-13
> **response to Reviewer g2YZ**
>
> 1. “The proposed method, Gandalf, is rather straightforward. The core idea seems to be just applying normalization and some thresholding on top of the LCG constructed in ECLARE.”
>
> * Please note that applying normalization and thresholding is _NOT the core idea of the paper_. As in this one, there are two main things in most papers (i) the main idea, and (ii) implementation/proposed method. It seems that the reviewer, ignoring the former, is focussing on the implementation/method part of the paper and doesn’t appreciate its simplicity.
> * While the above may be a personal choice, we do not share this view for three reasons (i) As most existing approaches in this domain are algorithmic (SiameseXML, ECLARE, DECAF etc), _none of the methods in XMC_ have taken a data-centric approach, and as such exploiting the problem characteristics to find a suitable augmentation methodology is still very non-trivial, (ii) simplicity of the method/algorithm lends it a broader applicability as in our case that it can be used in conjunction with the above existing methods, (iii) furthermore, given that it leads to enormous improvements, its simplicity enables the overall approach more interpretable and debuggable.
> * Finally, the method looks simple _only with the benefit of hind-sight_. To start with, it is not obvious (i) why one should take a data-centric approach, (ii) which data augmentation method, especially when heuristics like synonym and bert output don’t work, (iii) how to construct the input-output pairs in the augmentation scheme, especially in absence of any existing augmentation methods in XMC and why taking the LCG approach for constructing outputs is a reasonable one, (iv) connection with Vicinal Risk Minimization.
>
> --------
>
> 2. “It is not clear to me how the Gandalf and LabelMix are related to each other. They seem to be two independent data augmentation tricks and I failed to see the necessity in introducing LabelMix in this paper.”
>
> Please note that our work is the _first data augmentation based approach in XMC_. Therefore, to compare against Gandalf, we propose a LabelMix as a strong data augmentation baseline for XMC motivated by mixup techniques. We also try more commonly used approaches in NLP such as bert-based synonym replacement and report results in Table 3. We have further also dedicated a section to discuss why the latter heuristic data augmentation techniques fail in short-text XMC in section 6.
>
> -------------
>
> 3. “In Section the authors claim that LCG is was better than Label co-occurrence matrix but both are not as good as Gandalf which adopts a heuristic thresholding after row-wise normalization. However, I didn't see any empirical study that compares these three settings. It would be more convincing if there are some label-to-label augmentation baselines in the comparison.”
>
> We believe that there is some misunderstanding in interpreting our remarks on LCG and Label co-occurrence matrix. Note that, purely statistical label co-occurrence matrix are highly unreliable in XMC (ProXML [1], Section 4). We thus propose to leverage LCG (as proposed in ECLARE) instead, which is _just a smoothed version_ of the label co-occurrence matrix, with an additional row-wise normalization to bring values between [0, 1]. To remove noisy correlations we put a threshold value of 0.1. We simply name the data augmentation technique as Gandalf.
>
> [1] Data scarcity, robustness and extreme multi-label classification, Rohit Babbar, and Bernhard Schölkopf, 2019

---

### Official Review · Reviewer_qcbk · 2022-10-26

**Confidence:** 3
**Correctness:** 3
**Technical Novelty And Significance:** 2
**Empirical Novelty And Significance:** 2
**Recommendation:** 3

**Clarity, Quality, Novelty And Reproducibility:**

**Clarity**
* The paper can be improved on their clarity on how their proposal is presented. It is quite unclear how the actual training objectives are defined (e.g. loss functions) and how the newly generated examples are integrated into the objective.
* Algorithm 1 should be written in pseudo-code for those who do not have prior Python knowledge.


**Strength And Weaknesses:**

**Strength**
* Since Gandalf is a pure data augmentation technique that can be applied to some set of algorithms easily. (however, it’s a bit arguable for some other sets of algorithms that can do similar techniques more organically. see weakness #1).
* Authors try to set up a reasonable data-augmentation baseline, LabelMix, influenced by the popular “Mix-up” technique.

**Weakness**
* Selecting k most confusing examples via label similarities is not novel. It is sometimes called “hard-negatives” that is widely adopted in this field (for example, DPR [1], RocketQA [2] and ANCE [3]). The main difference is to use dense label space or to use the label relationship graph (they are quite interchangeable, but the dense space has nicer properties). It is also partially observed in Table 3 as using the soft-labels from the label graph further drastically increases the performance over hard labels.
* Defining the label graph solely based on the label co-occurrences might have its limitations because it does not use any textual understanding of each label. Commonly, the dense label space is trained jointly with the actual retrieval task [1,2,3] and hence captures deeper understanding of the label to label similarities based on both bipartite co-occurrence relationships and textual understanding. Moreover, solely depending on co-occurrence relationships can be very noisy when we have large label space (e.g. thousands of millions or billions).
* The work was not evaluated against common dense retrieval algorithms such as [1,2,3].

[1] Karpukhin, Vladimir, et al. "Dense passage retrieval for open-domain question answering." arXiv preprint arXiv:2004.04906 (2020).
[2] Qu, Yingqi, et al. "RocketQA: An optimized training approach to dense passage retrieval for open-domain question answering." arXiv preprint arXiv:2010.08191 (2020).
[3] Xiong, Lee, et al. "Approximate nearest neighbor negative contrastive learning for dense text retrieval." arXiv preprint arXiv:2007.00808 (2020).


**Summary Of The Paper:**

Extreme classification (or deep information retrieval) has been a popular research field that matches an input text (query) to a label that often is also a text. This paper focuses on a sub-field where both the input text and the label text are short. The proposed training algorithm is a data augmentation algorithm that generates similar examples to the observation based on the label to label similarities. Authors claim this new data augmentation technique was able to improve significantly on benchmark datasets.


**Summary Of The Review:**

The paper does not convey a significant novelty in this field and was not sufficiently compared to many core techniques in this field.

-----
Post rebuttal response:

I thank authors to provide detailed feedback. I also read other reviewers' feedback and concluded this paper does require further development to contain further contribution in this field. I still concerns the novelty of this paper (similar to Reviewer g2YZ).

Here are some detailed points that the author replied.

> Gandalf is only a method to leverage Label-Text/Features as data points whose ground truth labels are unknown.

This actually limits the novelty of the method. Gandalf's novelty is limited in introducing a heuristic method to generate additional labels.

> Contrast to Hard Negative Mining

Thanks for the detailed description. I wonder then what's the actual benefit of using Galdalf's soft-labels. We often found strong negatives that are closer to the decision boundary helps achieving more discriminative models especially when the label space is large (XMC). Wouldn't it be more beneficial Galdalf to focus on closer but not positive labels?

---

> ### Author Response · Authors · 2022-11-13
> **responses to Reviewer qcbk**
>
> 1. On Hard Negative Mining and using dense label space for retrieval task.
>
> Please note that all methods where Gandalf has been tested, already use hard negative mining and mine hard negatives from the dense label space. Gandalf is only a method to leverage Label-Text/Features as data points whose ground truth labels are unknown. In fact, the soft targets generated through Gandalf are, in all approaches, clubbed with hard negatives selected from the dense label space and trained. A detailed explanation can be found in our next comment - [Contrast to Hard Negative Mining](https://openreview.net/forum?id=05ff9BRSMzE&noteId=9x0B5iNHcW).
> ________
> 2. "Moreover, solely depending on co-occurrence relationships can be very noisy when we have large label space"
>
> We agree with reviewers remark and would like to highlight that this is the reason why do not simply use a label co-occurrence matrix but instead choose to use a version of the same as proposed in ECLARE. The LCG as proposed in ECLARE alleviates this issue by skewing the vanilla label co-occurrence matrix in the favour of tail labels. To further curb the impact of noisy correlations, we apply a threshold on the values in the normalized LCG.
> ________
> 3. The work was not evaluated against common dense retrieval algorithms such as [1,2,3].
>
> We believe the reviewer has misjudged our field of work. Our work is dedicated to the Extreme Classification domain and not Dense Retrieval/Open Domain QA. We show the effectiveness of our proposed data augmentations on almost all state-of-the-art (short-text) extreme classifiers. Please check the [Extreme Classification Repository](http://manikvarma.org/downloads/XC/XMLRepository.html) for more details.
> ________
> 4. "It is quite unclear how the actual training objectives are defined (e.g. loss functions) and how the newly generated examples are integrated into the objective."
>
> For Gandalf, the training objectives remain the same as in the original approaches. Gandalf only proposes a method to generate the positive labels for a "label-feature when leveraged as a data point" as the ground truth labels are unknown. The label feature becomes the data point and positive labels generated via Gandalf. We shall make this clearer in an updated version of the paper.

---

> > ### Comment · Reviewer_qcbk · 2022-12-12
> > **Re: rebuttal**
> >
> > I thank authors to provide detailed feedback. I also read other reviewers' feedback and concluded this paper does require further development to contain further contribution in this field. I still concerns the novelty of this paper (similar to Reviewer g2YZ).
> >
> > Here are some detailed points that the author replied.
> >
> > > Gandalf is only a method to leverage Label-Text/Features as data points whose ground truth labels are unknown.
> >
> > This actually limits the novelty of the method. Gandalf's novelty is limited in introducing a heuristic method to generate additional labels.
> >
> > > Contrast to Hard Negative Mining
> >
> > Thanks for the detailed description. I wonder then what's the actual benefit of using Galdalf's soft-labels. We often found strong negatives that are closer to the decision boundary helps achieving more discriminative models especially when the label space is large (XMC). Wouldn't it be more beneficial Galdalf to focus on closer but not positive labels?

---

### Decision · Program_Chairs · 2023-01-20

**Decision:**

Reject

**Justification For Why Not Higher Score:**

The paper received rather too strict scores, but certainly it needs to be improved to eliminate any confusions. Therefore, it has to be rejected.

**Justification For Why Not Lower Score:**

N/A

**Metareview: Summary, Strengths And Weaknesses:**

The paper introduces a new approach for extreme multi-label classification based on data augmentation. The authors considers a specific instance of the problem where objects are short texts and the labels are associated with short textual descriptions. The experimental results are promising.

The underlying idea is simple, but it does not mean that it should not be considered for publication. Nevertheless, the reviewers had found several problems and three of them recommended to reject the paper. The authors were able to defend some of the critical comments. However, the consistency of the ratings shows that the paper needs to improve in the presentation layer in order to eliminate any confusions.

The authors should make it more clear how the method is different from the existing ones, what is their contribution and what is the relation between both methods introduced in the paper. Also the empirical studies could be improved as the current ones seem to be designed to favor the introduced method.

On the other hand it might be that the simple augmentation approach is the best one can do for this specific problem. In such a case, the paper would need to be positioned differently. It should rather discuss the problem than promote a method.